# Structural Features and Immunomodulatory Effects of Water-Extractable Polysaccharides from *Macrolepiota procera* (Scop.) Singer

**DOI:** 10.3390/jof8080848

**Published:** 2022-08-13

**Authors:** Yordan Nikolaev Georgiev, Ondrej Vasicek, Balik Dzhambazov, Tsvetelina Georgieva Batsalova, Petko Nedyalkov Denev, Lili Ivaylova Dobreva, Svetla Trifonova Danova, Svetlana Dimitrova Simova, Christian Winther Wold, Manol Hristov Ognyanov, Berit Smestad Paulsen, Albert Ivanov Krastanov

**Affiliations:** 1Laboratory of Biologically Active Substances, Institute of Organic Chemistry with Centre of Phytochemistry, Bulgarian Academy of Sciences, 139 Ruski Blvd., 4000 Plovdiv, Bulgaria; 2Department of Biophysics of Immune System, Institute of Biophysics, Czech Academy of Sciences, 135 Kralovopolska, 612 65 Brno, Czech Republic; 3Department of Developmental Biology, Plovdiv University Paisii Hilendarski, 24 Tsar Assen Str., 4000 Plovdiv, Bulgaria; 4Department of General Microbiology, The Stephan Angeloff Institute of Microbiology, Bulgarian Academy of Sciences, 26 Acad. Georgi Bonchev Str., 1113 Sofia, Bulgaria; 5Bulgarian NMR Centre, Institute of Organic Chemistry with Centre of Phytochemistry, Bulgarian Academy of Sciences, 9 Acad. Georgi Bonchev Str., 1113 Sofia, Bulgaria; 6Department of Pharmacy, University of Oslo, P.O. Box 1068 Blindern, 0316 Oslo, Norway; 7Department of Biotechnology, University of Food Technologies, 26 Maritza Blvd., 4002 Plovdiv, Bulgaria

**Keywords:** *Macrolepiota procera*, polysaccharides, NMR, immunomodulatory activity, inflammation, prebiotic activity, *Clostridium beijerinckii*, probiotic bacteria, biofilm, *Escherichia coli*

## Abstract

*Macrolepiota procera* (MP) is an edible mushroom used in the treatment of diabetes, hypertension and inflammation. However, the structure and biological effects of its polysaccharides (PSs) are unclear. This study investigates the structural features of a PS complex from MP (MP-PSC), its immunomodulatory activities and effects on probiotic and pathogenic bacteria. MP-PSC was obtained by boiling water, and PSs were characterized by 2D NMR spectroscopy. The immunomodulatory effects on blood and derived neutrophils, other leukocytes, and murine macrophages were studied by flow cytometry, chemiluminescence, spectrophotometry, and ELISA. The total carbohydrate content of MP-PSC was 74.2%, with glycogen occupying 36.7%, followed by *β*-D-glucan, *α*-L-fuco-2-(1,6)-D-galactan, and *β*-D-glucomannan. MP-PSC (200 μg/mL) increased the number of CD14+ monocyte cells in the blood, after ex vivo incubation for 24 h. It dose-dependently (50–200 μg/mL) activated the spontaneous oxidative burst of whole blood phagocytes, NO, and interleukin 6 productions in RAW264.7 cells. MP-PSC exhibited a low antioxidant activity and failed to suppress the oxidative burst and NO generation, induced by inflammatory agents. It (2.0%, *w*/*v*) stimulated probiotic co-cultures and hindered the growth and biofilm development of *Escherichia coli*, *Streptococcus mutans* and *Salmonella enterica*. MP PSs can be included in synbiotics to test their immunostimulating effects on compromised immune systems and gut health.

## 1. Introduction

*Macrolepiota procera* (Scop.) Singer (MP), commonly called the parasol mushroom, is a basidiomycete fungus of the Agaricaceae family, which is highly appreciated due to its taste qualities. The mushroom alimentary use has been documented by the local people living in the Jirisan and Lombard Stelvio National Parks in South Korea and Italy, respectively, and in the Balkans [1,2,3]. It is a source of proteins, polyunsaturated fatty acids, tocopherols, indoles, minerals (incl. Se), fiber, carotenoids, phenolics, triterpenoids, alkaloids, etc. [4,5,6,7,8,9,10,11]. It has been recommended that the freezing of the fruiting bodies of MP is a better technique to preserve the nutritional quality of the mushroom than canning [12]. The health risk from the accumulation of heavy metals and rare elements in wild mushrooms, influenced by environmental and biological factors, has also been assessed [13,14,15]. 

The parasol mushroom is used in folk medicine for the treatment of diabetes and hypertension, and its decoctions are applied in inflammations in the oral cavity [5,16]. The antidiabetic effect of MP can be partially confirmed by the inhibitory activities of its hexane and methanolic extracts on *α*-amylase and *α*-glucosidase [17,18]. The anti-inflammatory effect of a mushroom methanolic extract can be linked to the in vitro inhibition of acetylcholinesterase and butyrylcholinesterase enzymes [18]. MP extracts with organic solvents have shown other in vitro actions related to inflammatory and infectious processes, such as antibacterial, antifungal, antitumoricidal, antiviral, antiplatelet, and antineutrophil aggregation activities [8,14,18,19,20,21,22]. The anti-inflammatory activity has partly been attributed to some lanostane-type triterpenoids and cinnamic acid, isolated from the mushroom, inhibiting in vitro NO production in phagocytes [20,23]. Zara et al. have found that an 80% ethanolic extract of MP exhibits anticancer activity against A549 lung adenocarcinoma cells with an IC_50_ = 6.18 µg/mL, after 24 h of treatment [24]. The antitumor effect of this mushroom extract has been explained by its inhibitory activity on glucose-6-phosphate dehydrogenase enzyme, which is important for cancer progression. Previously, Seçme et al. have determined that a MP extract suppresses the expression of cyclin genes, regulating cell progression, and Bcl-2, inhibiting apoptosis, while it activates genes involved in triggering apoptosis, such ascaspase-3, -9, Bax and the genome protective tumor suppressor gene p53 [19]. Additionally, silver nanoparticles, containing a MP extract, have suppressed gene and protein expressions of HSP27, HSP70, and HSP90 in A549 cells [25].

Not only mushroom extracts, obtained with organic solvents, have shown favorable pharmacological activities. Interestingly, an aqueous extract of MP has expressed growth-inhibitory and antibiofilm activities against *S*. *infantis*. The antibacterial effect was partly explained by the inhibition of L-amino acid oxidase, without further elucidation of the active compounds [26]. Doh-Hee et al., provoked by the alimentary use of MP, have found that an aqueous extract (80–2000 μg, i.v.) manifests in vivo immunostimulating activity and inhibits metastasis in colon26-M3.1 tumor-bearing mice [27].

A limited number of the bioactive water-soluble molecules in MP have been identified in aqueous extracts. A new *β*-trefoil lectin, isolated from the parasol mushroom, has been associated with its in vitro antiparasitic activity [28]. Cysteine protease inhibitors, named macrocypins, suppressing the growth of Colorado potato beetle larvae have also been found in the parasol mushroom [29]. In fact, edible mushrooms are a source of specific ribonucleases with potent insecticidal, antimicrobial, and antitumor properties, and of bioactive peptides with antibiotic, antithrombotic, and antihypertensive effects [30,31,32]. Apart from that, Kim et al. have demonstrated that the antitumor effect of a water-soluble PS–protein conjugate, isolated from a submerged culture of *Lepiota procera*, is expressed by its immunopotentiation activity, but not via a direct cytotoxic effect against sarcoma 180 tumors in ICR mice [33]. The promising in vivo antitumor effects of mushroom PS(–protein) complexes have already been reviewed, and PSs, such as schizophyllan, lentinan, and krestin (PS–protein complex), are used as anticancer and immunomodulatory agents in Asia [34]. These complexes combine the beneficial effects of both PSs and proteins. Furthermore, it has been found that MP mycelium PSs show protective activity on experimental male mice against nonylphenol toxicity via alleviating the oxidative stress, apoptosis, autophagy, and inflammatory responses in testicular cells [35]. Water-extractable PSs from the wild parasol mushroom have exhibited antioxidant and prebiotic activities in vitro, but their structure and immunomodulatory effects have not been elucidated [36,37]. Interestingly, the chemical profile, health-promoting effects, production, conservation, and use of the parasol mushroom as food have been reviewed, but information on the structural features and immunomodulating activity of its PSs is missing [38]. It is not known whether the chemical composition and immunomodulatory activity of PSs obtained from submerged cultures of MP differ from those of the wild mushroom, which is widely consumed. Furthermore, it was hypothesized that the PSs in the wild parasol mushroom are partly responsible for the documented immunostimulating, anti-inflammatory and antimicrobial effects of its aqueous extracts and decoctions. Therefore, the present study aimed to characterize the chemical features of a water-extractable PS complex (PSC) from the fruiting bodies of MP, its immunomodulatory activities on different leukocytes, and its effects on the development and biofilm formation of probiotic and pathogenic bacteria. It was suggested that this will help in the evaluation of the potential for MP PSs to be tested in synbiotic foods to potentiate innate immunity, supporting the treatment of immune-related and infectious diseases.

## 2. Materials and Methods

### 2.1. Chemicals, Consumables and Raw Material

All reagents were obtained from Sigma-Aldrich (St. Louis, MO, USA), and the organic solvents and consumables were delivered by VWR International (Radnor, PA, USA) if not indicated otherwise. The fruiting bodies of MP were collected in October 2018 near the town of Svoge (Western Bulgaria) and delivered dried, with a certificate of origin №3/2018/DG001, by the company Beatrix-77 Ltd., (Novi Iskar, Bulgaria). 

### 2.2. Preparative Methods 

#### 2.2.1. Alcohol-Insoluble Solids (AIS)

The fruiting bodies (100 g) of MP were ground to powder and defatted with petroleum ether (40–60 °C, 1:10), for 2 h at room temperature (RT). The extract was filtered through a nylon cloth and the residue was exhaustively treated with petroleum ether overnight. The recovered solids were extracted with 80% (*v*/*v*) ethanol (1:10) for 1 h at 65 °C and left overnight at RT. The solid residues were extracted twice with 80% ethanol (1:10) for 2 h at RT. The AIS were additionally treated for 1 h with 100% acetone (1:10) and air-dried. 

#### 2.2.2. Water-Extractable PSC

Fifty grams of AIS from MP fruiting bodies was subjected to a double extraction with boiling ultrapure water (0.055 µS/cm, Adrona Crystal B6.1, Riga, Latvia) for 1 h, employing a ratio of AIS to water of 1:28. The extract was filtered through a nylon cloth and the residue was washed with ultrapure water (150 mL). Then, the extract with the washing water was centrifuged at 4000× *g* for 25 min at RT. The supernatant was filtered with a Büchner funnel (KA 4 paper filters, Papírna Perštejn s.r.o., Perštejn, Czech Republic) and concentrated 4 times in a rotary vacuum evaporator at 40 °C. The concentrated extract was centrifuged, filtered again and the PSs were coagulated with 95% (*v*/*v*) cold ethanol (1:4) at 4 °C overnight. The PSs were recovered by centrifugation (4000× *g* for 30 min at 4 °C), dissolved in ultrapure water and dialyzed (MWCO 12–14 kDa, VISKING^®^, SERVA Electrophoresis GmbH, Heidelberg, Germany) against deionized water for 72 h at 4 °C. Finally, PSs were centrifuged at RT to remove any formed insoluble solids, filtered and freeze-dried. The obtained PSC from MP was named MP-PSC. 

### 2.3. Analytical Methods

#### 2.3.1. General Methods

MP-PSC (5 mg) and AIS of MP (7 mg) were dispersed in 72% (*w*/*w*) H_2_SO_4_ and pre-hydrolyzed for 1 h at 30 °C in a shaking water bath (NUVE, Asagi Ovecler Ankara, Turkey). They were diluted with ultrapure water (1 mg/mL) to 1 M H_2_SO_4_, and further hydrolyzed at 100 °C for 3 h in a block heater (SBH200D, Stuart^®^, Staffordshire, UK). The total uronic acid content in the hydrolyzed samples was assayed by the 3-phenylphenol method in a Skalar San^++^ autoanalyzer (Analytical BV, Breda, The Netherlands), using GalA as a calibration standard [39]. The absorbance was read at 530 nm and the results were presented as the anhydrouronic acid content (AUAC). The total carbohydrate content in the hydrolyzed MP-PSC and AIS of MP was analyzed by the phenol-sulfuric acid method of Dubois et al., using Glc for the preparation of a standard curve [40]. The absorbance measurement was performed at 490 nm. The total content of glycogen in the PSC (100 mg) was assayed using the Glc oxidase (*Aspergillus niger*)/peroxidase (horseradish)/4-aminoantipyrine method, after depolymerization of glycogen into Glc by a combined use of heat-stable *α*-amylase (*Bacillus licheniformis*) and amyloglucosidase (*A*. *niger*) [41]. The presence of methoxyl esters in the PSC (1–10 mg/mL) was examined by the enzymatic-colorimetric method of Anthon and Barrett, using methanol as a standard [42]. The acetyl content of MP-PSC (10 mg/mL) was evaluated by the hydroxamic acid reaction method, as described by McComb and McCready, using *β*-D-glucose pentaacetate for construction of a calibration curve [43]. The total protein content of the sample and AIS of MP (100 mg) was determined by the Kjeldahl method [44]. Nitrogen content was estimated as ammonia liberated from the samples, and it was determined by the acetylacetone–formaldehyde method, using ammonium sulfate for the construction of a calibration curve [45]. A conversion factor of 5.61 was applied to calculate the results [46]. Additionally, the protein content of MP-PSC (2 mg/mL) was determined by the Coomassie^®^ brilliant blue G-250 (Abcam, Cambridge, UK) staining method of Bradford, using bovine serum albumin as a standard [47]. The total phenolic content in the PSC (3 mg/mL) was analyzed according to Singleton and Rossi, using ferulic acid for the preparation of a calibration curve [48]. The protein and phenolic contents were determined directly in MP-PSC without prior protein or phenolic extraction. The complex was assayed in duplicate, and the results were expressed as the mean ± standard deviation (SD) (n = 6).

#### 2.3.2. Determination of Monosaccharide Composition 

The monosaccharide composition of MP-PSC (2 mg) was analyzed using the method of Chambers and Clamp, as described by Nyman et al. [49,50]. The sample, with added mannitol (100 µg) as an internal standard, was hydrolyzed with anhydrous 3 M HCl in methanol at 80 °C for 24 h. The hydrolyzed sample was dried with the help of a stream of nitrogen, and the formed methyl glycosides of the monosaccharides released were converted into their respective trimethylsilyl derivatives. The amino sugars were transformed into their re-N-acetyl derivatives with acetic anhydride and pyridine, as described by Wold et al. [51]. The derivatized monosaccharides were assayed on a Trace™ 1300 GC (Thermo Fisher Scientific™, Waltham, MA, USA), equipped with a flame-ionization detector and Restek Rtx-5 silica column (30 m length, 0.25 mm i.d., 0.25 μm film thickness, Restek Corporation, Bellefonte, PA, USA), in a scan mode (40–450 m/z). The carrier gas was helium (99.999%) at a constant pressure mode (0.7 bar). The injection volume was 1 µL (split, 1:10), and the injector and detector temperatures were set at 250 °C and 300 °C, respectively. The initial oven temperature was 140 °C, and then it was increased at 1 °C/min to 150 °C, held for 3 min, followed by an increase of 3 °C/min to 170 °C, then held for 5 min and finally increased at 15 °C/min to 310 °C, at which it was kept for 3 min. The derivatized monomers were elucidated by comparing the retention times of unknown compounds with those of analytical-grade standard (Glc, Gal, Man, Rha, Fuc, Ara, Xyl, Rib, GlcA, GalA and GlcNAc). The results were analyzed using Chromeleon software, version 6.80 (Dionex Corporation, Sunnyvale, CA, USA).

The presence of 3-deoxy-D-*manno*-2-octulosonic acid (Kdo) in MP-PSC (3 mg) was qualitatively determined according to the periodate thiobarbituric acid method [52,53]. A RG-II-containing PS from *Lavandula angustifolia* was used as a positive control [54].

#### 2.3.3. Determination of Molecular Weight Distribution

The molecular weight distribution of the PSs in MP-PSC (3 mg/mL) was determined using an Agilent 1220 system (Agilent Technology, Santa Clara, CA, USA), coupled with a Bio SEC-3 column (4.6 × 300 mm, 300Å, 3 μm, Agilent) and a refractive index detector 1260 RID. The mobile phase was 150mM NaH_2_PO_4_ (pH = 7.0), operating at a flow rate of 0.5 mL/min at RT. The sample injection volume was 20 μL and the detector temperature was 40 °C. Two samples of MP-PSC (3 mg/mL) were dissolved in water for 24 h, then they were centrifuged at 18,187× *g* for 10 min at 20 °C prior to analysis. Pullulan standards (Shodex standard P-82 kit, Showa Denko-K.K., Kawasaki, Japan), with molecular weights ranging from 0.59 × 10^4^ to 78.8 × 10^4^ g/mol, were used for the preparation of a standard curve by plotting the log10 value of the molecular weight versus retention time. The results were analyzed using EZChrom Elite Compact software, version 3.3.2. (Agilent Technology, Santa Clara, CA, USA). The weight-average (Mw) and number-average (Mn) molecular weights and polydispersity index (Mw/Mn) of MP-PSC were calculated.

#### 2.3.4. Microbial Lipopolysaccharides

The existence of lipopolysaccharides (LPS) in MP-PSC (2 mg) was checked by means of the detection of 3-hydroxy fatty acids, as their characteristic constituents in the form of acetylated fatty acid methyl esters, on a GC-EI/MS-QP 2010 system (Shimadzu, Kyoto, Japan), equipped with a Restek Rxi-5MS column (30 m length, 0.25 mm i.d., 0.25 m film thickness, Restek Corporation, Bellefonte, PA, USA) in a SIM mode. LPS of *E. coli* (serotype O111:B4) was used as a calibration standard, and the intensity of the ion with 257 *m*/*z*, obtained from the acetylated ester of 3-hydroxy tetradecanoic acid, was used for the calculations [51,55]. The injection volume was 1 μL (split ratio of 1:10). The injector temperature was 250 °C, and the oven was set at 50 °C, then held for 2 min, heated to 90 °C (20 °C/min) and held for 1 min, then heated to 280 °C with a step of 5 °C/min, and held for 2 min. Data were analyzed by GC–MS solution software, version 2.10 (Shimadzu, Kyoto, Japan).

#### 2.3.5. Fourier Transform Infrared Spectroscopy (FTIR)

The FTIR spectrum of MP-PSC (2 mg) was acquired between 500 and 4000 cm^−1^, using the attenuated total reflectance technique (ATR) on a Nicolet^TM^ Avatar spectrometer (Thermo Fisher Scientific, Waltham, MA, USA), controlled by OMNIC 3.2. software (Thermo Fisher Scientific^TM^, Waltham, MA, USA). The spectrum was examined in Spectragryph software (Friedrich Menges, Oberstdorf, Germany). 

#### 2.3.6. Ultraviolet-Visible (UV-Vis) Spectroscopy

The UV-Vis spectrum of MP-PSC (1 mg/mL) was recorded between 190 and 700 nm on an LLG-uniSPEC 2 spectrophotometer (Lab Logistics Group GmbH, Meckenheim, Germany) using LLG-uniSPEC 1/2 software, cat. №6.263 609 (LLG, Meckenheim, Germany). 

#### 2.3.7. Nuclear Magnetic Resonance (NMR) Spectroscopy

MP-PSC (30 mg) was dissolved in 700 μL of D_2_O and lyophilized for a hydrogen–deuterium exchange. Then, it was re-dissolved in 700 μL of D_2_O and transferred with a Pasteur pipette into a NMR tube (D400-5-8, Deutero GmbH, Kastellaun, Germany), and analyzed at 60 °C on a Bruker Avance II^+^ 600 spectrometer (Bruker Biospin GmbH, Rheinstetten, Germany). ^1^H, ^13^C, and ^13^C/^1^H heteronuclear single quantum coherence (HSQC), ^1^H/^1^H correlation spectroscopy (COSY), decoupling in the presence of scalar interactions (DIPSI), rotating-frame nuclear Overhauser effect spectroscopy (ROESY), and ^13^C/^1^H heteronuclear multiple bond correlation (HMBC) spectra were obtained. Sodium 4,4-dimethyl-4-silapentane-sulfonate (DSS) served as an internal standard. The results were analyzed using Bruker TopSpin^TM^ software, version 4.0.7. (Bruker Biospin GmbH, Rheinstetten, Germany).

### 2.4. Biological Methods

#### 2.4.1. Flow Cytometric Analysis of Human Leukocytes

Venous blood was obtained from the cubital vein of healthy human volunteers (n = 4, 2 women, and 2 men, 25–36 age) in BD Vacutainer^®^ K2EDTA test tubes (Becton, Dickinson and Company, Franklin Lakes, NJ, USA) after overnight fasting. The major hematological parameters in the blood of the participants were also studied prior to analysis to confirm that they had no signs of a cold or related diseases. The blood samples were centrifuged at 1500× *g* for 15 min at RT, and the plasma was removed. The lysis of the red blood cells was performed with 0.84% NH_4_Cl solution, and the remaining cells were rinsed twice with phosphate-buffered saline (PBS). The white blood cells were pooled and centrifuged at 1000× *g* for 10 min. After that, the pellet was resuspended in Dulbecco’s modified Eagle’s medium (DMEM) (Gibco, Invitrogen, Waltham, MA, USA), supplemented with 10% fetal bovine serum (FBS), and stabilized antibiotic antimycotic solution (Merck KGaA, Darmstadt, Germany). The cell suspension was divided for a duplicate experiment. Then, the cells were transferred into a 12-well microplate (Techno Plastic Products AG, Trasadingen, Switzerland) at a density of 1 × 10^6^ cells/mL, and they were treated with MP-PSC (200 μg/mL) for 24 h. The treated cells were cultured in a humidified incubator at 37 °C and with a supply of 5% CO_2_. Leukocytes, grown in DMEM only, served as a control. After 24 h, the control and MP-PSC-treated groups were obtained by centrifugation, and the cells were resuspended in PBS, containing 5% FBS and 0.05% NaN_3_. The control and treated cell groups were stained with fluorochrome-labeled antibodies for CD3, CD4, CD16, CD19, CD64, and CD69 surface markers (BD Pharmingen™, BD Biosciences, Franklin Lakes, NJ, USA), and they were incubated for 10 min in the dark. Control and MP-PSC-stimulated cells were rinsed twice with a buffer and analyzed on a flow cytometer Cytomics FC500 (Beckman Coulter, Brea, CA, USA). The cell counts of different white blood cell populations in the control and treated groups were compared. Data were calculated as the mean percentages ± SD (n = 3). 

#### 2.4.2. Reactive Oxygen Species (ROS) Production from Whole Blood Phagocytes (WBP) and Isolated Neutrophils

Blood was obtained from the cubital vein of healthy individuals (age 18–50) in test tubes, using sodium citrate (0.38%) as an anticoagulant. For neutrophil isolation, the blood was gently mixed with 3% dextran and left at RT for 30 min. The obtained buffy coat was carefully overlaid on Histopaque 1077 and centrifuged at 390× *g* for 30 min at RT. The removal of the remaining erythrocytes was achieved by lysis with water. The neutrophils were rinsed with cold PBS and centrifuged at 190× *g* for 10 min, and then resuspended again in cold PBS. The neutrophil cell viability was checked on a CASY cytometer (Roche, Basel, Switzerland), and only cells with viability above 95% were used. The kinetics of ROS production by WBP or isolated neutrophils was determined by luminol-enhanced chemiluminescence (CL) for 90 min at 37 °C, using 96-well white flat-bottomed plates on an Infinite M200 spectrometer (Tecan, Männedorf, Switzerland) [56]. Spontaneous response: 25 μL of ten-times diluted blood or neutrophil cell suspension (2.5 × 10^5^ cells/well) in Hank’s buffered salt solution (HBSS) were mixed with 25 μL of 10 mM luminol (in 0.2 M borate buffer) and 25 μL MP-PSC (50, 100, and 200 μg/mL), dissolved in HBSS. The CL was measured immediately after a volume correction with HBSS to 250 μL. Inflammatory response: 25 μL of ten-times diluted blood or neutrophil cell suspension, mixed with 25 μL of 10 mM luminol and 25 μL MP-PSC (50, 100 and 200 μg/mL), were incubated with 25 μL of 62.5 μg/mL opsonized zymosan particles (OZP) or 25 μL of 0.81 μM phorbol 12-myristate 13-acetate (PMA). The volume was again adjusted with HBSS to 250 μL. The integral values of CL were used. The results were determined as a percentage of the control (without or with PMA/OZP) and expressed as the mean ± standard error of the mean (SEM) (n = 3) from three separate runs.

#### 2.4.3. Nitric Oxide (NO) and Interleukin 6 (IL-6) Productions from RAW264.7 Cells

The formation of nitrites in the culture medium of macrophages, stimulated with MP-PSC, was quantified using the Griess reagent (Abcam, Cambridge, UK) [57]. The nitrites served as an indirect indicator for NO production. RAW264.7 cells (TIB-71, American Type Culture Collection, Manassas, VA, USA) were cultured in 12-well flat-bottom plates at a cell density of 2.5 × 10^5^ cells/well with 100 µL of MP-PSC (50, 100, and 200 μg/mL), dissolved in DMEM-high Glc growth medium, containing 10% (*v*/*v*) FBS, and 1% (*v*/*v*) penicillin/streptomycin (HyClone^TM^, Cytiva, Marlborough, MA, USA). The incubation was performed for 24 h at 37 °C and with a CO_2_ supply of 5%. The final reaction volume was 300 μL. Additionally, the macrophages were co-treated with MP-PSC and 10 ng/mL LPS (*E. coli*/026:B6) under the same conditions. After 24 h, the supernatants were collected by centrifugation at 16,000× *g* at 4 °C for 5 min. Eighty microliters of each sample was mixed with 80 μL of the Griess reagent, and the reaction mixtures were incubated in 96-well plates at RT for 30 min in the dark. Absorbance measurement was performed at 540 nm (SPECTRA Sunrise^TM^ microplate reader, Tecan, Männedorf, Switzerland). For the preparation of a calibration standard curve, 0–52 μM NaNO_2_ solutions, prepared in DMEM, were used. Data were converted to a percentage of the control or LPS control, and they were presented as the mean ± SEM (n = 3) from three separate runs. 

Simultaneously, IL-6 production by RAW264.7 cells, treated with MP-PSC, LPS and MP-PSC+LPS, was also determined after 24 h of exposure. The analysis was performed according to an immunoassay kit Mouse IL-6 DuoSet (R&D Systems, Minneapolis, MN, USA). Data from three independent runs were converted to a percentage of the control (mean ± SEM, n = 3). 

#### 2.4.4. Cell Viability of RAW264.7 Cells

The possible cytotoxic effects of MP-PSC were checked on RAW264.7 cells after the treatment described in Section 2.4.3, as explained in [58]. The supernatants were replaced by a 200 μL 3-(4,5-dimethylthiazol-2-yl)-2,4-diphenyltetrazolium bromide (MTT) solution (0.125 mg/mL) and the samples were incubated for 1 h at 37 °C with a supply of 5% CO_2_. The MTT-containing medium was discarded, the plates were rinsed with deionized water, and then 300 μL of 10% Triton X-100 (in 0.1 M HCl) was added. The plates were incubated for 15 min on a shaker to dissolve the formazan crystals within the cells. Then, the absorbance was measured at 570 nm (SPECTRA Sunrise^TM^ microplate reader, Tecan, Männedorf, Switzerland). The results from three separate runs were converted to a percentage of the control (mean ± SEM, n = 3).

#### 2.4.5. In Vitro Antioxidant Activity

The oxygen radical absorbance capacity (ORAC) activity of MP-PSC was measured on a microplate reader FLUOstar OPTIMA (BMG Labtech, Offenburg, Germany) with an excitation at λ = 485 nm and emission at λ = 520 nm [59,60]. Fluorescein (FL) served as a fluorescent agent, and Trolox was used as a calibration standard. The antioxidant activity of MP-PSC was studied by comparing the area under the fluorescence curve relative to this area in the absence of the sample (control). MP-PSC was assayed in duplicate, and the results were calculated in micromole Trolox equivalents (TE) per gram of sample (mean ± SD, n = 6).

The hydroxyl radical averting capacity (HORAC) activity of the PSC was measured with an excitation at λ = 485 nm and emission at λ = 520 nm, according to [61]. FL was used as a fluorescent agent, and gallic acid served as a standard. The antioxidant activity of the sample was determined as explained for the ORAC method. The complex was assayed in duplicate, and the results were presented in micromole gallic acid equivalents (μmol GAE) per gram of sample (mean ± SD, n = 6). 

#### 2.4.6. Prebiotic, Growth-Inhibitory and Antibiofilm Activities

In a preliminary screening, the prebiotic activity of MP-PSC (2%, *w*/*v*) was studied on 44 different potential probiotic strains of *Lactobacillus delbrueckii* subsp. *bulgaricus*, *Lactiplantibacillus plantarum*, *Lacticaseibacillus rhamnosus*, *Limosilactobacillus fermentum* and some non-defined strains of *Lactobacillaceae*. They were isolated from Bulgarian dairy products (yogurt, katak, white-brined cheese, kashkaval, green cheese) and from products of human origin (including from breast milk). Then, the stimulatory effects of MP-PSC on *Clostridium beijerinckii* strains (4.3A and 6A), isolated from spontaneously fermented chickpea beans [62], and a newly isolated *Lactobacillus* sp. ZK9, from a sauerkraut fermentation, were investigated. The growth-inhibitory and antibiofilm formation activities of MP-PSC were examined on 7 pathogens: *Escherichia coli* HB 101 (K12), *Streptococcus mutans* DSMZ20523 (Leibniz Institute DSMZ-German Collection of Microorganisms and Cell Cultures GmbH, Braunschweig, Germany), *E. coli* 420, clinical *E. coli* ATCC 25922, *E. coli* ATCC 11775 (serovar O1:K1:H7), *E. coli* MG1655 (ATCC, Manassas, VA, USA), and *Salmonella enterica* subsp. *enterica* serovar Typhimurium WDCM 00031 (Vitroids™, Merck KGaA, Darmstadt, Germany). 

The two *C. beijerinckii* strains and *Lactobacillus* sp. ZK9 were grown in De Man, Rogosa, and Sharpe (MRS, Merck KGaA, Darmstadt, Germany) medium for 24 h at 37 °C. The bacterial cells were collected by centrifugation at 9000× *g* for 5 min at RT and washed with 0.85% NaCl to remove residual carbohydrates. Then, they were resuspended in a modified MRS medium, where Glc (carbon source) was replaced by MP-PSC, serving as a sole carbon source. The pre-washed cultures of the two *C. beijerinckii* strains in an exponential phase (20 h) alone or with each other, and in individual combinations with *Lactobacillus* sp. ZK9, were grown in microplates (96 wells, Cellstar^®^, Greiner Bio-One, Kremsmünster, Austria) at a cell density of 10^6^ CFU/mL (2 × 10^5^ cells/well) in the modified medium containing MP-PSC (2%, *w*/*v*). Lactose (2% *w*/*v*) added to the modified MRS broth served as a positive control. The aqueous solution of MP-PSC (2%, *w*/*v*) was sterile filtered through a 0.20 µm syringe filter (Corning^®^, Kaiserslautern, Germany) and aseptically added to the growth medium. The cell growth was studied at 570 nm on an INNO spectrophotometer (LTek, Gyeonggi-do, South Korea) after treatment at 37 °C for 0, 24, 48, 72, and 96 h. The experiments were performed in triplicate, and the results for the cell growth were presented as relative optical density (OD) units (mean ± SD, n = 3).

The pathogens were pre-cultured in a Brain Heart Infusion broth (Becton, Dickinson and Company, Difco™, Franklin Lakes, NJ, USA). After about 20 h, the cells in an exponential phase were collected by centrifugation. Then, the pre-washed cultures were inoculated in a modified MRS medium (pH 6.8) with MP-PSC (2%, *w*/*v*), serving as a unique carbon source. The treated cells were cultured for 24, 47, 72 and 96 h at 37 °C. The experiments were performed in triplicate for all strains. The cell growth inhibition by MP-PSC was calculated by the following formula: Inhibition (%) = 100 − A_sample_/A_control_ × 100, where A is the absorbance read. The results were presented as the mean % inhibition of the control ± SD (n = 3). 

After 96 h of incubation of the bacterial pathogens with MP-PSC, the biofilm development was determined by the crystal violet (CV) assay [63]. 

### 2.5. Statistical Analysis

The data obtained using the methods presented in Section 2.4.1 and Section 2.4.6 were analyzed by Statview software, version 5.0 (SAS Institute, Cary, NC, USA), using the one-sample Student’s *t*-test to compare the mean values ± SD to those from the (activated) controls. Similarly, the results obtained using the methods presented in Section 2.4.2, Section 2.4.3 and Section 2.4.4 were analyzed by Prism-6.01 software (GraphPad software, San Diego, CA, USA), applying the Bonferroni correction of *p* values for multiple comparisons (* *p*< 0.05).

## 3. Results

### 3.1. Primary Chemical Characterization of MP-PSC

The yield of dried AIS, obtained from MP, was 57.3% (*w*/*w*). The total protein (Kjeldahl method) and PS contents in the dried fruiting bodies of MP were 20.2% and 18.6%, respectively. The AUAC in dried MP was 1.3%, and it was calculated that AUAC in MP-PSC occupied 7% of the total PS content. 

The yield and results of the primary chemical analyses of the PSC from MP are shown in Table 1. The yield of MP-PSC was relatively low (2.9%). A lower extractability of water-soluble PSs from MP can be expected because of the rigid cell walls, containing chitin, which increases its structural strength. The total carbohydrate content of MP-PSC was 74.1 ± 0.7% (*w*/*w*). The glycogen content in MP-PSC was high, occupying 36.7% (*w*/*w*) of the total carbohydrate content. 

MP-PSC mainly contained neutral PSs, because the main monosaccharide in the complex was Glc (62.3% *w*/*w*) (Table 1). This was consistent with the results for glycogen content and AUAC in the complex and dried mushroom. MP-PSC most probably also had other linear and/or branched glucan-type PSs. In addition to Glc, the sample was rich in Gal (19.7%), and the less common 3-*O*-Me-Gal (2.7%) was also detected. The presence of Man (6.9%) and Fuc (3.4%) in MP-PSC proposed that Glc, Gal, Man, and Fuc may build some interesting heteropolysaccharide(s). Furthermore, monosaccharide composition data revealed that GalA and GlcA were the detected uronic acids in the complex. It was assumed that they were not the main building blocks in the mushroom PSs. The other charged monomer in MP-PSC was GlcNAc, which was present in a negligible amount, and it could originate from residual chitin linked to some of the water-soluble polymers extracted. Xyl was also detected in negligible amounts. The rare monosaccharide Kdo, which can be found mainly in some complex PSs from plant and microbial origin, was not detected in MP-PSC. The low acetyl content (0.3%) and the absence of methanol in MP-PSC revealed that some PS fractions could contain acetyl esters, but they were not methyl-esterified.

The sample contained several molecular populations (Appendix A). That is why it was named PSC but not directly PS, because it was composed of PSs and lower quantities of noncarbohydrate molecules (Table 1). The determined Mn of MP-PSC was 37.8 × 10^4^ g/mol, and its Mw was 66.3 × 10^4^ g/mol. The polydispersity index of the PSs in the complex was 1.8, which was in agreement with its molecular heterogeneity. The sample was characterized by a narrow molecular-weight distribution and a large share of fractions with a high molecular weight. The major molecular population in MP-PSC was 67.1 × 10^4^ g/mol and it occupied 62.4% from the elution profile of the PSs (Appendix A).

Some other compounds were also detected in the sample (Table 1). The total phenolics content in MP-PSC was 1.7% (*w*/*w*), which shows that they are contained in small amounts. The total protein content in the complex was determined independently by the Kjeldahl (17.5%) and Bradford (12.7%) methods, pointing out that the sample contained a considerable amount of proteins. The aqueous extract before precipitation of the PSs during the extraction was not deproteinized, because there was a risk of removal of possible PS–protein conjugates, which could show immunomodulating activity. The existence of noncarbohydrate constituents in MP-PSC was confirmed during the analysis of the UV region of its UV-Vis spectrum (Appendix A).

### 3.2. FTIR Analysis

Figure 1 represents the FTIR spectrum of MP-PSC, with designations of the bands. The spectrum contained signals typical for PSs. The bands at 3304 and 2921 cm^−1^ were assigned for *ν*(OH) and *ν*(CH) (CH_2_), respectively [64]. The peak at 2361 cm^−1^ most probably indicated absorbed atmospheric CO_2_. The absence of peaks at around 1600 and 1730 cm^−1^ confirmed that MP-PSC was poor in uronic acids (Table 1) and their methyl esters [64], and that the fatty acid content in the AIS was very low. Two important distinctive absorption bands were detected in the spectrum of MP-PSC at 1640 cm^−1^ and 1548 cm^−1^, which corresponded to *ν*(CO) (amide I) and *δ*(NH) and *ν*(CN) (amide II) of the contained proteins [65]. A negligible amount of GlcNAc was found, and no other amino sugars were detected in MP-PSC, and thus the band at 1548 cm^−1^ could not be due to amino sugars (Table 1). The water signal appears at around 1645 cm^−1^, but the presence of the other peak indicated that it was an absorption band arising from proteins. The area between 1408 and 1444 cm^−1^ could be a result of the *δs*(CH_2_) and *δs*(CH_3_) of proteins in MP-PSC [65]. The signal at 1241 cm^−1^ could indicate the presence of amide III (*δ*(N-H) and *ν*(C-N)), and acetyl esters in MP-PSC, which was in agreement with the colorimetric analyses [64,66]. 

The fingerprint region between 1200 and 1000 cm^−1^ contains the following important vibrations of the ring skeleton: CO, CC, C-OH, and COC of the glycosidic bonds [67]. It was confirmed that MP-PSC contained *α*-glucans (1149, 1020, 928 and 765 cm^−1^), a predominant proportion of which must originate from glycogen (Table 1) [68]. MP-PSC most likely contained *β*-glucans, as there was an absorption band for the *β*-glycosidic bond (COC) at 1077 cm^−1^ [69], which was further investigated during the 2D NMR analysis. 

### 3.3. NMR Structural Analysis

The major PS fractions in MP-PSC were analyzed by 2D NMR spectroscopy. Figure 2 shows the HSQC spectrum of MP-PSC and the results of the structural analysis are summarized in Table 2. Additionally, the ^1^H, ^13^C, and HMBC spectra of MP-PSC are presented in Appendix A, respectively. It was found that Glc was bound in an *α*-1,4-D-glucan and neutral branched (acetylated) *β*-D-glucomannan. From the intensities for the anomeric protons, it was concluded that the second PS was contained in smaller amounts. Intense through-bond resonances between H1 (5.36 ppm), C2 (74.4 ppm), C3 (76.0 ppm), C4 (80.1 ppm), and C5 (74.1 ppm) of *α*-D-Glc*p* were observed in the HMBC spectrum (Appendix A), proving the presence of α-1,4-D-glucan (glycogen) [70,71]. The following fragments were also identified: 1,4-*β*-D-Glc*p*, 1,4-*β*-D-Man*p*, 1,4,6-*β*-D-Man*p*, and *β*-D-Man*p* [72,73]. Based on inter-glycosidic interactions in the HMBC spectrum, it was found that *β*-1,4-D-Glc*p* and *β*-1,4,6-D-Man*p* were linked together by 1,4-glycosidic bonds [72]. A cross-peak was observed between C1 of *β*-1,4-D-Glc*p* and H4 of *β*-1,4,6-D-Man*p* at *δ*105.7/3.84 ppm in the HMBC spectrum. This was confirmed by the through-space internuclear spin–spin couplings at 4.52/3.84, 4.80/3.73(5), and 4.80/3.63 ppm for H1/H4 correlation between *β*-1,4-D-Glc*p* and *β*-1,4,6-D-Man*p*, H1/H4(3) and H1/H5 interactions of *β*-1,4,6-D-Man*p* and *β*-1,4-D-Glc*p* in the ROESY spectrum. No definite interactions were found in the 2D spectra to prove that *β*-1,4,6-D-Man*p* was replaced at *O*-6 by Man or the presence of 1,4-linked *β*-D-homomanan due to the low Man content. Both types of binding exist in glucomannans [72].

The acetyl group-specific couplings between H and C from the methyl group at 2.16–2.21/23.3 ppm were registered in the HSQC spectrum (Figure 2). The CO signal from the acetate resonating at 176.2 ppm was elucidated from the HMBC spectrum. The acetyl ester content in the sample was low, which agreed with the data in Table 1. The acetylation of Glc and Gal in MP-PSC was rejected, as it would lead to downfield chemical shifts, and it would be recorded in the HMBC and ROESY spectra. It was hypothesized that Man in glucomannan was *O*-2 and/or *O*-3 acetylated, as demonstrated by Makarova et al. [73].

A high amount of Gal (19.7%) and small amounts of 3-*O*-Me-Gal (2.7%) and Fuc (3.4%) were detected in MP-PSC. The existence of a terminal *α*-L-Fuc*p* was proved by the C1/H1, C5/H5, and C6/H6 cross-peaks in the HSQC spectrum at 104.1/5.07, 72.0/4.18 and 18.3/1.24 ppm (Figure 2), as well as by the intra-residue interactions between H1/H2 (5.07/3.84 ppm) and H5/H6 (4.18/1.24 ppm) within the ring in the COSY spectrum. In the HSQC spectrum, an interaction between the signals at 59.0 and 3.44 ppm was detected, which showed the presence of −OCH_3_ groups [74]. The substitution of *O*-3 with the methyl group at 1,6-*α*-D-Gal*p* was demonstrated by intra-residue resonances between -OCH_3_ at 3.44 ppm and H3 (3.53 ppm), H2 (3.89 ppm), or H4 (4.27 ppm) of Gal in the ROESY spectrum. Couplings were found between −OCH_3_ and H3 of *α*-1,6-(3-*O*-Me)-D-Gal*p* at 59.0/3.53 ppm or between C3 and −OCH_3_ at 81.7/3.44 ppm in the HMBC spectrum. C3 (81.7 ppm) of *O*-3 methylated α-1,6-D-Gal*p* showed intense interactions with H1 (4.98 ppm), H2 (3.89 ppm), and H4 (4.27 ppm) in the HMBC spectrum (Appendix A). The NMR analysis revealed that Gal and Fuc formed a heteropolymer in which *α*-D-Gal*p* and its methylated derivative were linked by 1,6-glycosidic bonds, and they were partially substituted at *O*-2 by *α*-L-Fuc*p* units, as reported by Komura et al. [74]. The existence of 1,6-*α*-D-Gal*p* was proved by the intense through-bond resonances between H1 (C1) and C6 (H6, 6′) or C5 (H5) at 4.98 (100.7)/69.4 (3.70, 3.89) or 4.98 (100.7)/71.7 (4.18) ppm in the HMBC spectrum [75]. This was confirmed by the H1/H6, 6′; (H1/H4), and H1/H5 through-space internuclear spin–spin couplings at 4.98/3.70, 3.89; (4.98/4.02) and 4.98/4.18 ppm in the ROESY spectrum. The existence of the fragment →6)-*α*-D-Gal*p*-2-*O*-*α*-L-Fuc*p*-(1→ in MP-PSC was revealed by the following inter-residue cross-peaks in the HMBC spectrum: H1 of *α*-L-Fuc*p* with C2 of *α*-1,2,6-D-Gal*p* at 5.07/80.8 ppm, H1 of *α*-1,6-D-Gal*p* with C6 of *α*-1,2,6-D-Gal*p* at 4.98/69.4 ppm [75]. The couplings between H1 of *α*-L-Fuc*p* and H2 of *α*-1,2,6-D-Gal*p* at 5.07/3.84 ppm and the low-intense cross-peak for H1/H6 of α-1,2,6-D-Gal*p* at 5.13/3.70 ppm in the ROESY spectrum supported the *O*-2 substitution with Fuc of 1,6-linked D-Gal*p*. The ratio between the integral intensities of H1 of *α*-1,6-D-Gal*p* (4.98 ppm) and H1 of *α*-1,2,6-D-Gal*p* (5.13 ppm) was 81:19, which revealed that about 19% of Gal was *O*-2 substituted. According to monosaccharide composition data, 19% of the Gal content in MP-PSC was calculated as 3.7%, which agreed with the Fuc content in the sample (3.4%). Considering that the ratio between the integral intensities of H1 of *α*-1,2,6-D-Gal*p* (5.13 ppm) and H1 of *α*-L-Fuc*p* (5.07 ppm) was 62:38, it indicated that not only Fuc was attached to *O*-2 of Gal. An additional analysis after further chromatographic fractionation of MP-PSC will unambiguously prove or reject the existence of a covalent binding between, e.g., fucogalactan and glucomannan fragments elucidated in the MP-PSC.

Peaks resonating in the range 0.8–3.0 ppm, which are characteristic for the protons of the N-H and N-CH_3_ groups of proteins, were found in the proton spectra of MP-PSC. This was in agreement with the protein content in the complex (Table 1). Protein structures were not further investigated because this was beyond the scope of the current study, which was focused on PSs. 

### 3.4. Ex Vivo Effects on Human Leukocytes

In order to test the effects of MP-PSC on different human white blood cells, flow cytometric analysis was performed (Figure 3). The results help us to find the possible cell type targets of MP PSs. MP-PSC most significantly increased the primary peripheral blood CD14+ monocyte cell levels (*p* < 0.01) (Figure 3A). The stimulating activity of the complex on granulocytes was very low and statistically insignificant (Figure 3B). In general, it was suggested from the results obtained with monocytes that MP PSs could serve as antigens for innate immunocompetent cells, such as macrophages and dendritic cells, which can originate from monocytes. MP-PSC did not influence CD4+ T cell levels (Figure 3C). MP-PSC expressed a suppressive activity on CD19+ B cells (Figure 3D) and activated CD69+ cell population (*p* < 0.05) (Figure 3E). The latter involves activated T lymphocytes, NK cells, but also B cells. Therefore, the mushroom PSs did not express any stimulatory activities on T and B lymphocytes under the studied conditions.

Furthermore, the oxidative burst from WBP and derived neutrophils treated with MP-PSC were investigated (Figure 4 and Figure 5). This was necessary to confirm the activation ability of the mushroom PSs on human innate immune cells. MP-PSC dose-dependently stimulated (50–200 μg/mL) the spontaneous ROS generation from WBP (*p* < 0.01) (Figure 4A). The effect observed for the sample was in complete agreement with the results for CD14+ cells and granulocytes (Figure 3A,B). Neutrophils were suggested to be the major contributors to the ROS production in the blood, and thus it was supposed that the PSs in MP could activate them. 

Additionally, a co-treatment with PMA or OZP was performed to test the anti-inflammatory potential of MP-PSC, because these molecules are famous inflammatory agents. When PMA or OZP was added, MP-PSC did not express an additive or synergic activity towards the ROS formation in WBP (Figure 4B,C). However, MP-PSC did not decrease the ROS generation after co-stimulation with OZP or PMA, and thus it did not express any in vitro anti-inflammatory-like behavior.

MP-PSC did not activate the spontaneous oxidative burst by blood-derived neutrophils (Figure 5A). This effect was not expected because MP-PSC did not decrease the cell count of granulocytes (Figure 3B). It was hypothesized that the blood environment and its cells, producing antibodies, different signaling molecules, and especially the complement were involved in the activation of neutrophils in the blood by MP-PSC. Similarly to the results with WBP, the sample did not clearly suppress the inflammatory activities of OZP and PMA, but an inverse dose–response relationship was found (Figure 5B,C). Although this effect was not statistically significant, it was suggested that MP-PSC could contain some compounds that counteract the inflammatory processes induced by PMA and OZP. 

### 3.5. In Vitro Effects on Murine Macrophages

MP-PSC could affect the monocyte and partly the granulocyte cell population counts (Figure 3A,B), which provoked further study of the ROS production in WBP and derived neutrophils treated with the sample (Figure 4A and Figure 5A). Logically, it was necessary to check the stimulating ability of the mushroom PSC on macrophages. The cell viability, NO and IL-6 productions of RAW264.7 cells treated with MP-PSC were tested (Figure 6). MP-PSC (50–200 μg/mL), and LPS control (10 ng/mL) inhibited the cell viability after 24 h of incubation (Figure 6A, *p* < 0.05). Similarly to the experiments on the ROS production, the possible anti-inflammatory activity of MP-PSC on murine macrophages was investigated. For that purpose, a simulated sterile bacterial infection with LPS in cells treated with MP-PSC was performed, and the survival rate of RAW264.7 cells under co-stimulation conditions was also studied (Figure 6B). Moreover, MP-PSC treatment suppressed the cells again, and thus the complex did not express any protective effects against LPS on the cell viability.

Importantly, it was confirmed that the dose of LPS administered induced the production of NO and IL-6 in RAW264.7 cells (Figure 6C,E), which proved that the endotoxin shock was successfully simulated. MP-PSC considerably induced the spontaneous production of NO (*p* < 0.01) and IL-6 production (*p* < 0.01) dose-dependently. Only activated macrophages produce NO and IL-6, and thus, the results probably confirmed that MP-PSC could activate innate immune cells. Co-treatment with MP-PSC and LPS led to an increased NO production (Figure 6D), which caused an additive effect on the inflammatory action of LPS. Therefore, it was concluded that MP-PSC did not express any anti-inflammatory activity against the endotoxin shock in murine macrophages under the studied conditions. Furthermore, the inverse dose–response relationship that was observed for the treatment with MP-PSC and PMA or OZP on WBP and derived neutrophils (Figure 4 and Figure 5) was not found for the NO production of macrophages incubated with MP-PSC and LPS. This implied the need for the determination of the possible LPS contamination of MP-PSC.

### 3.6. In Vitro Antioxidant Activity

Additionally, an investigation of the antioxidant activity of MP-PSC was performed. This was necessary to elucidate the possible influence of antioxidant constituents in MP-PSC on the production of ROS and NO by WBP, derived neutrophils, and murine macrophages, respectively. The ORAC activity of MP-PSC was 313.3 ± 23.9 μmol TE/g PSC on a dry weight basis. The complex did not express any antioxidant activity through the HORAC method. This could be explained by its composition and particularly by the low uronic acid content, because uronides exhibit metal-chelating properties. The ORAC method revealed that MP-PSC expressed a low antioxidant activity. The ORAC activity was in agreement with the low content of phenolic compounds in the sample (1.7%), which was achieved by the successful treatment of the fruiting bodies with organic solvents before the extraction of PSs. Thus, the antioxidants in MP-PSC did not play a major role in the documented effects of the sample in the treatment of human white blood cells and upon spontaneous production and that co-stimulated with PMA, OZP, or LPS of ROS and NO, respectively.

### 3.7. Effects on Lactic Acid Bacteria and Pathogenic Bacteria

It was suggested that MP-PSC could express prebiotic effects because it contained neutral Glc-based polymers, which are traditionally known as prebiotics. In fact, probiotic bacteria support the immune system in the gastrointestinal tract and produce growth-inhibitory metabolites against pathogens, implying that the prebiotic and antibacterial activities of MP-PSC should be tested. The results from the preliminary screening with 44 different microorganisms revealed that PSs in MP-PSC did not stimulate the cell growth and biofilm development of all *Lactobacillus* strains after 96 h of cultivation under the investigated conditions (not shown). This could be explained by the complex composition and structural features of the macromolecules in MP-PSC—mainly PSs and some proteins, which cannot be degraded by their intrinsic enzyme machinery. 

The possible prebiotic effect of the mushroom PSC was investigated in mixed cultures between *C. beijerinckii* (4.3A and 6A) strains and *Lactobacillus* sp. ZK9 (Figure 7). Clostridia were chosen because they can digest lignocellulosic biomass in nature. However, MP-PSC did not stimulate the bacterial growth of both strains of clostridia when they were individually treated with the complex, serving as a sole carbon source. In fact, there was no cell growth when *Lactobacillus* sp. ZK9 was incubated with MP-PSC (data not shown). After that, both clostridia were co-cultured, and negligible cell growth was observed (Figure 7). However, a mixed culture between *Lactobacillus* sp. ZK9 and each clostridium led to considerable assimilation of the mushroom PSs, which was statistically significant. The cell growth was comparable to that on lactose (2%, *w*/*v*). Therefore, a symbiotic growth-stimulatory activity was found, and this phenomenon could be explained by the better catabolic properties of both microbes living together and their cross-feeding. *C. beijerinckii* was proposed to release simpler sugars from MP-PSC that were fermented by *Lactobacillus* sp. ZK9 to lactic acid, which was converted into butyric acid by the former gaining ATP.

Apart from the possible prebiotic effect, the growth-inhibitory action of MP-PSC was tested against gut-associated microbial pathogens. The complex substantially suppressed the growth of *E. coli* 25922, *E. coli* MG1655, *E. coli* 420, *E. coli* 11775, *Str*. *mutans* and *S*. *enterica* between 47–72 h (Figure 8). For example, MP-PSC reduced the growth of *Str. mutans*, which generally causes tooth decay, with 54.6 ± 4.6%, 67.0 ± 2.8%, and 52.4 ± 1.0% after 24, 47, and 96 h of treatment, respectively. Interestingly, the growth reduction for *S. enterica* was higher between 24 and 47 h, but after that, the inhibition was decreased. This could be explained by a possible metabolic adaptation of the bacteria to assimilate MP-PSC, which was not observed for the other pathogens in the same manner. Additionally, MP-PSC suppressed the biofilm formation of *Str. mutans* and other pathogenic strains after 96 h of treatment (Appendix A). Pathogens formed biofilms on lactose (2%) as a sole carbon source (positive control). It is important to further investigate the dose-related response for the antibacterial activity of MP-PSC because the effect was achieved at a relatively high concentration in comparison with most of the known antimicrobial organic classes of low-molecular weight compounds. Additionally, it should be noted that the dose–response relationship for MP PSs is also needed for the evaluation of their practical use in the development of multifunctional nutritional and healthcare-related products.

## 4. Discussion

### 4.1. Influence of Noncarbohydrate Compounds on the Immunomodulatory Effects of MP-PSC

In general, the activation ability of MP-PSC on various human leukocytes was not very prominent. This suggested that the mixture of glycogen, *α*/*β*-glucan, glucomannan, and fucogalactan did not remarkably affect the activity of immune cells under the studied conditions. The influence of noncarbohydrate molecules (25.9%) in MP-PSC on the effects observed for its activity on different human leukocytes, oxidative burst, and murine macrophage activation should not be ignored. The UV-Vis spectrum of MP-PSC revealed the existence of UV-absorbing noncarbohydrate constituents in the complex (Appendix A), as some of them were elucidated in Section 3.1 (Table 1). In particular, proteins were the predominant noncarbohydrate compounds that could affect the immunomodulatory activity of PSs in MP-PSC. In fact, a bioactive PS–protein conjugate has been isolated from the mycelium of *L. procera*, but the role of the protein(s) in the expression of activity has not been elucidated [33]. The influence of the protein part on the biological activity of the whole protein–PS conjugate/complex is still not a regular subject of study during the evaluation of immunomodulatory activity (including molecular mechanisms of action) of mushroom PSs. It has been demonstrated that a PS–protein complex CQNP from *Chenopodium quinoa* induces lower iNOS expression and NO production in RAW264.7 cells than the deproteinized PS sample D-CQNP [76]. The activation of NF-*κ*B and MAPK signaling pathways by D-CQNP is similar to that of LPS. Additionally, the content of phenolics and the related ORAC antioxidant activity of MP-PSC were low, and thus the phenolics were not considered important contributors to immunomodulatory activity. The FTIR analysis also showed that MP-PSC did not contain detectable amounts of fatty acids, which are normally found in the lipid fraction of the mushroom, and they can affect the immune-related bioassays.

On the other hand, some of the biological results provoked the determination of possible LPS contamination of MP-PSC. It was found that the sample was contaminated with LPS (3.8%), therefore, the studied immunomodulatory effects of the complex presented in Section 3.4 and Section 3.5 can be compromised by the contained LPS. Furthermore, LPS triggers apoptosis in macrophages through an autocrine synthesis of TNF-α [77], and therefore the inhibitory effect on the cell viability of macrophages for the LPS-contaminated MP-PSC was expected. Similar conclusions could be suggested for the inhibitory effect on CD19+ B cells (Figure 3D) and the activated CD69+ cell population (Figure 3E). The high levels of NO and IL-6 in macrophages treated with MP-PSC could be partly explained by the contamination of the sample with LPS. However, mushroom *β*-glucans and fucogalactans can activate murine RAW264.7 cells [78,79]. The presence of LPS in MP-PSC did not lead to the activation of neutrophils (Figure 5A), but it was potentially partially related to the oxidative burst during blood treatment (Figure 4A). This suggested a difference in the effect of microbial LPS on cell type and phagocyte origin, and/or on the complex effects of mushroom PSs on LPS response. MP-PSC did not exhibit anti-inflammatory behavior against PMA and OZP treatment of human blood and derived neutrophils (Figure 4 and Figure 5). Otherwise, mushroom fucogalactans can exhibit in vivo anti-inflammatory and antinociceptive activities [74]. Contamination of the sample did not allow a real assessment of the combinatorial immunomodulatory effects of the glucans, glucomannans, and fucogalactans contained in MP-PSC. However, estimation of LPS content helped to achieve a better understanding of the immunomodulatory activities expressed by the complex. LPS content analysis has to be included in the experimental design of studies aiming to evaluate the biological activity of natural compounds. Nowadays, many researchers still underestimate the serious impact of low concentrations of LPS on the expression and immunomodulatory activity of PSs.

### 4.2. Evaluation of MP as a Valuable Source of Immunomodulatory and Prebiotic PSs

MP has been recognized as a good source of soluble and insoluble dietary fibers. Its consumption can provide an essential amount of the recommended daily dietary allowance of fibers [4,80]. It has been estimated that the *β*-glucan content in MP is between 7.9 and 11.2% [4,81]. Interestingly, PSs from a submerged culture of MP have already been partly studied. Kim et al. have isolated from *L. procera* two antitumor heteromannoglucan conjugates with proteins, which contain representative amounts of Fuc and traces of Gal and Xyl [33]. They have not detected large amounts of Gal and have not mentioned the presence of its methylated analog in the isolated PSs. Therefore, it could be speculated that there was a difference in the PS composition between the submerged cultivated and wild mushroom (Table 1). The authors have not presented the glycosidic linkage types of monosaccharides in the PSs, which left that question open. Similarly to our study, Samanta et al. have obtained from the fruiting bodies of *M. dolichaula*, using boiling water, *β*-glucan (PS-I) and *α*-fucogalactan (PS-II) with immunostimulating properties on RAW264.7 cells, splenocytes and thymocytes in vitro [78,79]. This was in agreement with the stimulatory effects and dosages used of MP-PSC on human CD14^+^ monocytes, ROS, and NO productions from RAW264.7 cells and WBP, respectively. Interestingly, the backbone of the *β*-glucan has been composed of 1,6-linked D-Glc*p* with branches at *O*-3 of 1,3,4-linked D-Glc*p*. They have found a 1,4-*β*-D-Glc*p* structure, as in MP-PSC [78], but *α*-glucan or glucomannan has not been found, as in MP. This can be chiefly explained by species differences. Furthermore, the authors have demonstrated that PS-II exhibits in vitro antioxidant activity against hydroxyl radicals (EC_50_ = 875 μg/mL), superoxide radicals (EC_50_ = 80 μg/mL) and oxidation of *β*-carotene (EC_50_ = 345 μg/mL) [79]. The reported antioxidant activities of PS-II and the ORAC activity of MP-PSC (313.3 ± 23.9 μmol TE/g) revealed that PS-II and MP-PSC should not be considered potent antioxidant agents. However, no information on the presence of LPS, proteins, or phenolics in PS-I and PSI-II has been included, which is important for a better understanding of the PS contribution to biological activities. Additionally, Wang et al. have isolated with hot water, from another species of *Macrolepiota* genus, *M. albuminosa*, 3-*O*-methylated glucomannogalactan, glucomannogalactan and galactoglucan, containing proteins and exhibiting useful functional properties [82]. The authors have suggested that 1,2,6-linked *α*-D-Gal*p* is *O*-2 substituted by terminal *β*-D-Man*p* or 1,6-*β*-D-Glc*p*. Their work was in agreement with our hypothesis, based on the NMR study, that not only Fuc can be attached to *O*-2 of Gal in the fucogalactan fragment of MP-PSC. Interestingly, a 3-*O*-methylated mannofucogalactan, where ←1-*α*-L-Fuc*p*-3-*O*-*α*-D-Man*p* disaccharide is attached to *O*-2 of Gal has also been determined in *Grifola frondosa* [83]. Therefore, it cannot be excluded that fucogalactan and glucomannan fragments in MP-PSC may be covalently bound, which will be studied after chromatographic fractionation of MP-PSC. However, Wang et al. have not found any Fuc and α-glucans in *M. albuminosa* PSs [82], which should be accepted as a structural difference between the PSs of both species. For many immunomodulating mushroom *β*-glucans, most of the molecular mechanisms of immunomodulation are clear, along with the anticancer action, while different cell surface receptors have been identified [84]. Unlike *β*-glucans, immune-related receptors for immunomodulating fucogalactan recognition are still unclear, which, from a biochemical point of view, should be elucidated in the near future. For example, Li et al. have revealed that a sulfated fucogalactan from the seaweed *Undaria pinnatifida* can be recognized by the elastin-peptide receptor on monocytes, assuming the Gal-binding lectin site of the receptor interacts with Gal [85]. However, there are some structural differences between mushroom and seaweed fucogalactans, which can restrict the possibility to activate the same receptors on the cell surface of target cells.

Fucogalactan fragments in MP-PSC are not the only fragments that deserve discussion, because heteromannan fragments can also be involved in the immunomodulatory activity. Interestingly, the detected 1,4- and 1,4,6-linked *β*-D-Man*p* residues in the complex are not as common for mushroom and yeast PSs as they are for plants [72,73]. However, a PS (LVF-I), containing →4)-*β*-D-Man*p*-(1→ and →6)-*β*-D-Man*p*-(1→ units, with similar MP-PSC monosaccharide and glycosidic linkage compositions, has been isolated from *Lactarius volemus* Fr. [86]. LVF-I expresses in vitro anticancer and splenocyte stimulatory effects, but it also induces RAW264.7 cells to produce NO, IL-6 and TNF-*α* at an mRNA level, at doses of 250–1000 µg/mL. The authors have found that LVF-I acts in synergism with LPS on the proliferation of splenocytes. This again opens up the question of whether their sample has been contaminated by LPS, because the same effect was observed for MP-PSC on macrophages even at much lower PS concentrations (Figure 6D,E). Furthermore, a glucomannan composed of 1,2-linked *β*-D-Man*p* branched at *O*-6 with 1,3-linked *β*-D-Glc*p* units has been purified from *Agaricus blazei* and it shows in vivo antitumor activity against Sarcoma 180 solid tumors in ICR/SLC mice [87]. Interestingly, a neutral PS (NHSP), with in vivo hypoglycemic properties in mice with type 2 diabetes, has been isolated from *Hohenbuehelia serotina*, and it also contained Man*p* in a *β*-anomeric configuration (→6)- *β*-D-Man*p*-(1→) [88].

Apart from that, MP-PSC failed to stimulate the growth of any of the studied lactobacilli in a MRS medium, supplemented with 2% MP-PSC in a preliminary screening investigation. It was suggested that the native PSs in MP-PSC are not able to exhibit prebiotic activity without any structural modifications on the probiotic bacteria screened. However, Nowak et al. have succeeded in growing *L*. *acidophilus* ATCC 4356 and one of two *L. rhamnosus* strains of human origin in a MRS medium, supplemented with 1.5% (*w*/*v*) MP PS, instead of Glc [37]. In the current study, when a mixed culture of *C. beijerinckii* strains and *Lactobacillus* sp. ZK9 was studied, it was found that MP-PSC can be metabolized. Importantly, *C. beijerinckii* strains have been defined as acetic and butyric acid-positive producers, and they are normally found in the human gut [89]. *C. beijerinckii* is also included in commercial probiotic products. It is well-known that butyrate and other short-chain fatty acids are important for supporting gut health. The potential clinical application of butyrate-producing probiotic bacteria in conditions of dysbiosis, such as diarrhea, inflammatory bowel disease, Crohn’s disease, obesity, diabetes, and sepsis, has been reviewed [90]. Glc-containing polymers in MP-PSC were possibly metabolized by the co-cultures, because it has been reported that mushroom *α*- and *β*-glucans can serve as prebiotics [91].

Although MP-PSC did not exhibit suppressive effects on LPS-treated RAW264.7 cells, it reduced the growth of clinical and reference bacterial pathogens. Interestingly, an aqueous extract of MP has shown growth-inhibitory and antibiofilm activities against *S*. *infantis*, as the antibacterial effect has been partly explained by inhibition of L-amino acid oxidase [26]. For example, Strapáč et al. have reported that an aqueous extract of the parasol mushroom expresses antiquorum sensing activity against human clinical isolated *Pseudomonas aeruginosa* 119 and *P. aeruginosa* 44, which were obtained from bovine lungs [92]. They have suggested that phenolics in aqueous MP extracts are responsible for the antimicrobial activities observed, because they have determined the total phenolic content in the extracts. Although phenolics are well-known for their growth-inhibitory effects, there are plenty of other active molecules in the aqueous extracts. In particular, the results from the current study suggest that the water-extractable PS–protein complexes found in the aqueous extract are also involved in the antibacterial and antibiofilm properties. Suppression of the growth of different gut-associated pathogens implies a protective effect, as the potential activity of the mushroom PSs in this action is not completely understood. For example, Kumar et al. have demonstrated that *L. acidophilus s*timulates the function of monocarboxylate transporter 1 for colonic luminal absorption of butyric acid, which is impaired by *E. coli* intestinal infection [93]. Therefore, the detected cross-feeding phenomenon between *C. beijerinckii* strains and *Lactobacillus* sp. ZK9 grown on MP-PSC and the antibacterial activity of the complex against *E. coli* seem promising in the case of intestinal pathogens. Additionally, from a technological point of view, the aqueous mushroom extract, rich in PSs and other bioactive compounds, can find application as a functional food. For example, the addition of ethanolic extract of MP, rich in phenolic compounds and fatty acids, in cooked sausages increased their microbial stability during 30 days of storage [94].

The immunomodulatory effects, prebiotic and antibacterial activities of MP-PSC stimulate the future investigation of its PSs as functional additives in synbiotic foods to support the treatment of conditions of compromised immune health and gut health. Many mushroom PSs have already been tested in animal models and humans, and their beneficial effects on malignant diseases, diabetes, other metabolic disorders, inflammations and infections have been proven [34,95].

## 5. Conclusions

This study demonstrated that the water-extractable PSs in MP are involved in the immunomodulatory and antibacterial properties of its aqueous extracts known from ethnomedicine. MP-PSC differed from the PSs isolated from the submerged mushroom culture by its higher amounts of Gal and the identified 3-*O*-methylated Gal. In general, MP is distinguished from other species in the same genus with the accumulation of high amounts of water-soluble α-glucans and/or glycogen, and the presence of 1,4,6-*β*-D-Man*p* structures in its (hetero-)*β*-glucans. PSs in MP-PSC successfully activated the human innate immune response ex vivo by increasing the blood monocyte cell number and inducing spontaneous oxidative burst in WBP. Unlike some published data on the prebiotic activity of MP PSs, it was demonstrated that the mixture of *α*/*β*-glucans, fucogalactan and glucomannan fragments in MP-PSC stimulate the growth only of some probiotic co-cultures, but not that of 44 individual probiotic and candidate probiotic cultures. The detected ex vivo immunostimulating effects, the cross-feeding phenomenon between the butyrate-producing *C. beijerinckii* strains and *Lactobacillus* sp. ZK9, and the antibacterial activity of the complex in vitro, seem promising against intestinal pathogen infections. Therefore, the PSs of parasol mushroom deserve to be further studied in functional and synbiotic foods to support the treatment of immune-related and infectious diseases. In conclusion, chromatographic fractionation, protein removal and LPS decontamination of PSs in MP-PSC are needed for a complete evaluation of their individual immunomodulatory activities.

## Figures and Tables

**Figure 1 jof-08-00848-f001:**
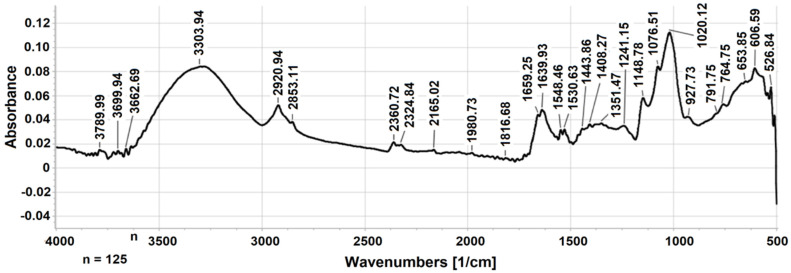
ATR-FTIR spectrum of MP-PSC.

**Figure 2 jof-08-00848-f002:**
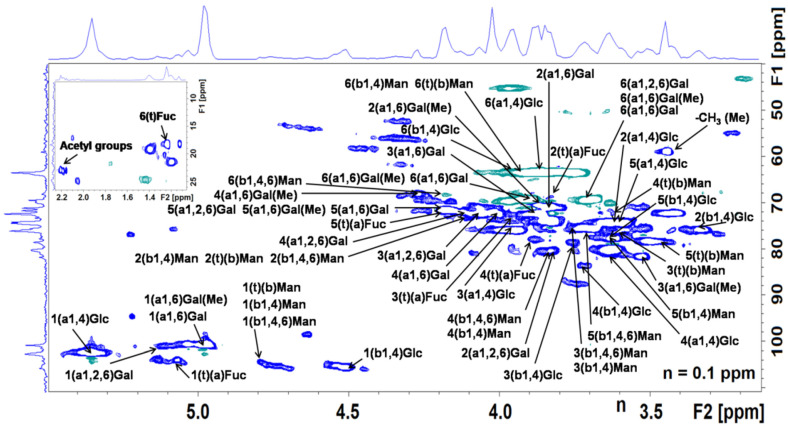
^1^H/^13^C HSQC spectrum of MP-PSC. Sodium 4,4-dimethyl-4-silapentane-sulfonate was used as an internal standard. Annotation: the first digit shows the proton or carbon number in the ring and *α* (a) and *β* (b) anomeric configurations, and glycosidic linkage types are added in the brackets. Abbreviations: Me (methyl group), t (terminal).

**Figure 3 jof-08-00848-f003:**
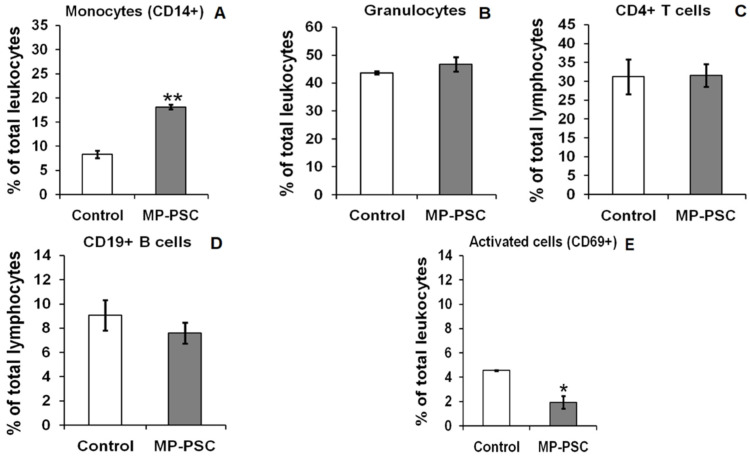
Ex vivo activity of MP-PSC (200 μg/mL) on: monocytes (**A**), granulocytes (**B**), CD4+ T cells (**C**), CD19+ B cells (**D**), and activated CD69+ cells (**E**), isolated from healthy human individuals, after incubation for 24 h. The asterisks indicate statistical significance (* *p* < 0.05, ** *p* < 0.01) vs. control—untreated cells.

**Figure 4 jof-08-00848-f004:**
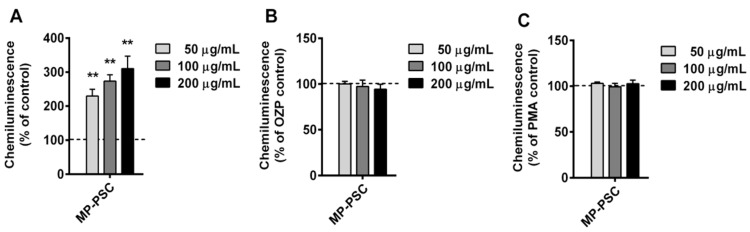
Ex vivo (**A**) spontaneous, (**B**) opsonized zymosan particle (OZP)-, and (**C**) phorbol myristate acetate (PMA)-activated reactive oxygen species (ROS) production from whole blood phagocytes treated with MP-PSC. Asterisks indicate statistical significance (** *p* < 0.01) against control (blood only, +OZP or +PMA).

**Figure 5 jof-08-00848-f005:**
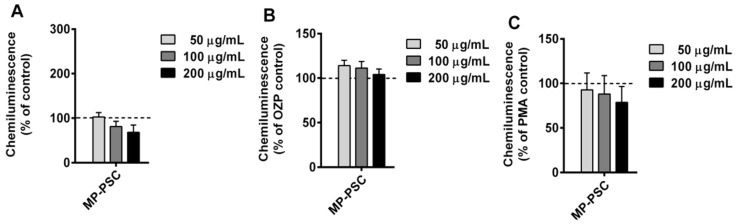
Ex vivo activity of MP-PSC on: (**A**) spontaneous, (**B**) OZP-, and (**C**) PMA-activated ROS production from isolated neutrophils.

**Figure 6 jof-08-00848-f006:**
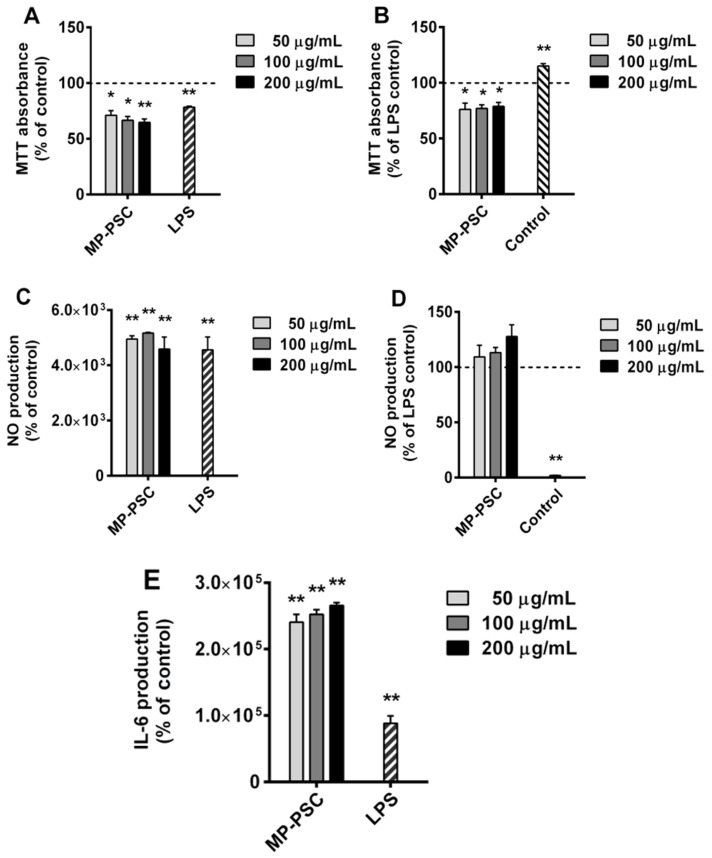
In vitro activities of MP-PSC on murine macrophages (RAW264.7 cells): (**A**) MTT cell viability test after treatment with MP-PSC; (**B**) MTT cell viability, after co-treatment with MP-PSC and lipopolysaccharides (LPS); nitric oxide (NO) synthesis, induced by MP-PSC alone (**C**) or in combination with LPS **(D**); (**E**) interleukin 6 (IL-6) synthesis, induced by MP-PSC alone. The asterisks indicate the statistical significance (* *p* < 0.05, ** *p* < 0.01) vs. control—untreated cells (growth medium or LPS-treated cells).

**Figure 7 jof-08-00848-f007:**
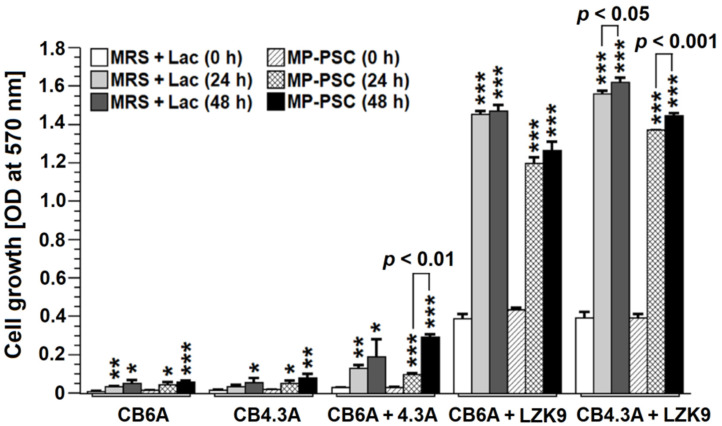
In vitro cell growth of *C. beijerinckii* 43A or 6A and co-cultures between 43A and 6A or with *Lactobacillus* sp. ZK9 cultivated in a modified MRS medium, containing MP-PSC (2%, *w*/*v*) as a sole carbon source. The asterisks indicate statistical significance (* *p* < 0.05, ** *p* < 0.01 and *** *p* < 0.001) vs. control—cells cultivated in a modified MRS broth with added lactose (Lac) (2%, *w*/*v*).

**Figure 8 jof-08-00848-f008:**
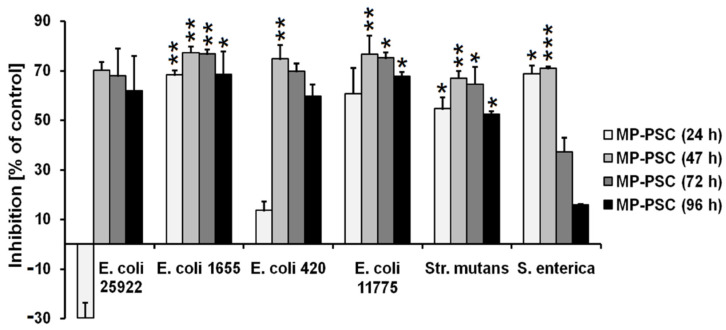
In vitro antibacterial activity of MP-PSC (2%, *w*/*v*) against clinical and reference pathogenic strains, cultured in a MRS medium without other carbon sources. The asterisks indicate statistical significance (* *p* < 0.05, ** *p* < 0.01 and *** *p* < 0.001) vs. control—cells cultivated in a modified MRS medium with added lactose (2%, *w*/*v*).

**Table 1 jof-08-00848-t001:** Yield and primary chemical characterization of a water-extractable polysaccharide complex isolated from the fruiting bodies of *M. procera* (MP-PSC). Data are expressed on a dry weight basis.

Parameters	MP-PSC
Yield in the dried fruiting bodies (%)	2.9
Total carbohydrate content (%)	74.1 ± 0.7
Total glycogen content (%)	27.2
AUAC (%) ^1^	1.8 ± 0.02
Methanol content (*w*/*w*%)	n.f. ^2^
Acetyl content (*w*/*w*%)	0.3 ± 0.005
Monosaccharide composition (*w*/*w*%) ^3^	
Fuc	3.4
Xyl	1.8
Man	6.9
Gal	19.7
Glc	62.3
GlcA	1.2
GalA	1.3
3-*O*-Me-Gal	2.7
GlcNAc	0.7
Kdo ^4^	n.f.
Relative Mw (g/mol) ^5^	66.3 × 10 ^4^
Total protein content (%)	17.5 ^6^ (12.7 ± 0.2) ^7^
Total phenolic content (%)	1.7 ± 0.02

^1^ Anhydrouronic acid content; ^2^ not found; ^3^ calculated on the basis of total carbohydrate content; ^4^ 3-deoxy-D-*manno*-2-octulosonic acid; ^5^ weight-average molecular weight; ^6^ Kjeldahl method; ^7^ Bradford method.

**Table 2 jof-08-00848-t002:** ^13^C and ^1^H NMR chemical shift assignment (in ppm) of MP-PSC, referenced to DSS ^1^.

Residue	C-1 H-1	C-2 H-2	C-3 H-3	C-4 H-4	C-5 H-5	C-6 H-6,6′	CH_3_O-
→1)-*α*-Glc*p*-(4→	102.6 5.36	74.4 3.63	76.0 3.95	80.1 3.63	74.1 3.61	63.4 3.87	-
→6)-*α*-Gal*p*-(1→	100.7 4.98	71.1 3.84	72.4 3.89	72.4 4.02	71.7 4.18	69.4 3.70; 3.89	-
→6)-3-*O*-Me-α-Gal*p*-(1→	100.7 4.98	70.4 3.89	81.7 3.53	68.1 4.27	71.7 4.18	69.4 3.70; 3.89	59.0 3.44
→2,6)-*α*-Gal*p*-(1→	101.4 5.13	80.8 3.84	72.9 4.06	72.0 4.10	71.7 4.18	69.4 3.70; 4.00	-
*α*-Fuc*p*-(1→	104.1 5.07	68.8 ^2^ 3.84	73.2 3.95	77.7 3.89	72.0 4.18	18.3 1.24	-
→4,6)-*β*-Man*p*-(1→	104.1 4.80	72.9 4.10	75.5 3.76	80.5 3.84	75.9 3.70	68.1 4.18	-
*β*-Man*p*-(1→	104.1 4.80	72.9 4.10	75.5 3.63	72.0 3.62	78.5 3.47	63.8 3.93	-
→4)-*β*-Man*p*-(1→	104.1 4.80	72.9 4.10	75.5 3.76	80.5 3.84	77.7 3.63	63.8 3.95	-
→4)-*β*-Glc*p*-(1→	105.7 4.52	76.0 3.33	78.5 3.75	83.9 3.73	77.7 3.63	63.4 3.95	-

^1^ DSS, sodium 4,4-dimethyl-4-silapentane-sulfonate; ^2^ elucidated from the HMBC spectrum.

## Data Availability

All data are available in the paper and in the Appendix A.

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
