# Peer review of "Structural Features and Immunomodulatory Effects of Water-Extractable Polysaccharides from Macrolepiota procera (Scop.) Singer"

_jof, 2022, doi:10.3390/jof8080848_

Round 1

Reviewer 1 Report

The manuscript entitled ‘Structural features and immunomodulatory effects of water-extractable polysaccharides from Macrolepiota procera (Scop.) Singer’ by Georgiev et al. reports the characterization and the chemical features of a water-extractable PS complex (PSC) from the fruiting bodies of MP and its immunomodulatory activities on different leukocytes. The manuscript is of interest and results are well discussed. I recommend the publication in Journal of Fungi after minor revisions as reported below.

-Mushrooms are a rich source of bioactive proteins/peptides (enzymes) with several potential antitumor and bioinsecticidal activities. Thus, I suggest to review the recent literature and add general information regarding the presence of bioactive proteins/peptides in mushrooms before talk about the subject of the study. See: https://doi.org/10.3390/toxins14020084; https://doi.org/10.3390/toxins13040263; https://doi.org/10.1016/j.foodchem.2022.133655;

-Page 1, line 46: use the full name of mushroom the first time it appears in introduction, then use the abbreviated scientific name;

-Page 4, line 165: which reference for the conversion factor used. Please add;

-Page 9, Table 1: The values refer to the fresh weight? Moreover, the authors compare total proteins obtained by Kjeldahl and Bradford method. For the latter method it is not clear the extraction method used. Please add more information.

-Page 16, line 646: g of what? Fresh or dry?

Author Response

Dear Ms. Ruby Shu and Reviewer 1,

We would like to thank you for the time spent on our manuscript, the valuable recommendations that improve the quality of our paper, and the positive evaluation. We place our responses on the Reviewers’ comments below. The Reviewers’ comments are put in quotes and our responses are colored in yellow. All changes in the manuscript are colored in red and highlighted by "Track Changes".

Reviewer 1

"The manuscript entitled ‘Structural features and immunomodulatory effects of water-extractable polysaccharides from Macrolepiota procera (Scop.) Singer’ by Georgiev et al. reports the characterization and the chemical features of a water-extractable PS complex (PSC) from the fruiting bodies of MP and its immunomodulatory activities on different leukocytes. The manuscript is of interest and results are well discussed. I recommend the publication in Journal of Fungi after minor revisions as reported below." – We would like to thank the Reviewer for the positive evaluation of our manuscript.

"- Mushrooms are a rich source of bioactive proteins/peptides (enzymes) with several potential antitumor and bioinsecticidal activities. Thus, I suggest to review the recent literature and add general information regarding the presence of bioactive proteins/peptides in mushrooms before talk about the subject of the study. See: https://doi.org/10.3390/toxins14020084; https://doi.org/10.3390/toxins13040263; https://doi.org/10.1016/j.foodchem.2022.133655;" - We have added the three new references dealing with bioactive proteins and peptides found in edible mushrooms, according to the Reviewers’ 1 recommendations. It was also included another reference (https://doi.org/10.1021/jf403615f) on cysteine protease inhibitors with insecticidal activity found in Macrolepiota procera. Please, read lines (86-90): Cysteine protease inhibitors, named macrocypins, suppressing the growth of Colorado potato beetle larvae have also been found in the parasol mushroom [29]. In fact, edible mushrooms are a source of specific ribonucleases with potent insecticidal, antimicrobial, and antitumour properties, and of bioactive peptides with antibiotic, antithrombotic, and antihypertensive effects [30-32].

"-Page 1, line 46: use the full name of mushroom the first time it appears in introduction, then use the abbreviated scientific name;" – We have eliminated the technical gap.

"-Page 4, line 165: which reference for the conversion factor used. Please add;" – We added a new reference (https://doi.org/10.1021/jf00096a011) to support the use of the conversion factor of 5.61 for calculation of total protein content in mushroom samples. Please, read line (171): A conversion factor of 5.61 was applied to calculate the results [46].

"-Page 9, Table 1: The values refer to the fresh weight? Moreover, the authors compare total proteins obtained by Kjeldahl and Bradford method. For the latter method it is not clear the extraction method used. Please add more information." – It was indicated that the results in Table 1 are on a dry weight basis. To avoid any confusion for determination of noncarbohydrate compounds in the polysaccharide complex, it was clarified that the total contents of these compounds were determined directly in the sample, without any further isolation. Please, read lines (176-177): The protein and phenolic contents were determined directly in MP-PSC without prior protein or phenolic extraction. We also included data for the total protein and anhydrouronic acid contents of the fruiting bodies of M. procera (see lines: 397-400) because these results were not added in the original submission. However, they are part of the chemical characterization of the initial source used for polysaccharide extraction.

"-Page 16, line 646: g of what? Fresh or dry?" - It was shown that ORAC antioxidant activity value in the text was calculated on a dry weight basis of the polysaccharide complex. Please, read lines (654-655): The ORAC activity of MP-PSC was 313.3±23.9 μmol TE/g PSC on a dry weight basis.

08.08.2022                              Sincerely yours,

Plovdiv, Bulgaria                     Yordan Georgiev - corresponding author

Reviewer 2 Report

Dear Editor, based on carefully reading the manuscript entitled "Structural features and immunomodulatory effects of water-extractable polysaccharides from Macrolepiota procera (Scop.) Singer", I came to the conclusion that it is an extremely attractive topic that is presented in a nice and concise way, although it contain an overview of an extremely extensive literature in the field in a wide time frame. The authors investigated the polysaccharides complex of Macrolepiota procera and its immunomodulatory and effect on probiotic and pathogenic bacteria. Numerous results data are presented in tables or figures, which makes it significantly easier to follow the text and complements it graphically. The text is written in excellent English. Based on everything stated, I am happy to propose that the paper be published in its current form.

Author Response

Dear Ms. Ruby Shu and Reviewer 2,

We would like to thank you for the time spent on our manuscript, and the positive evaluation. We place our response on the Reviewers’ comment below. The Reviewers’ comment is put in quotes and our response is colored in yellow. All changes in the manuscript are colored in red and highlighted by "Track Changes".

Reviewer 2

"Dear Editor, based on carefully reading the manuscript entitled "Structural features and immunomodulatory effects of water-extractable polysaccharides from Macrolepiota procera (Scop.) Singer", I came to the conclusion that it is an extremely attractive topic that is presented in a nice and concise way, although it contain an overview of an extremely extensive literature in the field in a wide time frame. The authors investigated the polysaccharides complex of Macrolepiota procera and its immunomodulatory and effect on probiotic and pathogenic bacteria. Numerous results data are presented in tables or figures, which makes it significantly easier to follow the text and complements it graphically. The text is written in excellent English. Based on everything stated, I am happy to propose that the paper be published in its current form." – We would like to thank the Reviewer for the positive evaluation of our manuscript.

08.08.2022                                    Sincerely yours,

Plovdiv, Bulgaria                           Yordan Georgiev - corresponding author